# The effect of nasal and oral breathing on airway collapsibility in patients with obstructive sleep apnea: Computational fluid dynamics analyses

Masaaki Suzuki [ID][1]☯*, Tadashi Tanuma[2]☯

1 Department of Otolaryngology, Teikyo University Chiba Medical Center, Chiba, Japan, 2 Laboratory of Fluid-Structural Simulation and Design, Strategic Innovation and Research Center, Teikyo University, Tokyo, Japan

☯ These authors contributed equally to this work.
* suzukima@med.teikyo-u.ac.jp

**Data Availability Statement:** All relevant data are within the paper.

**Funding:** There were no funding or sources of support received during this study.

## Abstract

### Objective

The purpose of this study was to investigate the effect of breathing route on the collapsibility of the pharyngeal airway in patients with obstructive sleep apnea by using computational fluid dynamics technology.

### Methods

This study examined Japanese men with obstructive sleep apnea. Computed tomography scans of the nose and pharynx were taken during nasal breathing with closed mouth, nasal breathing with open mouth, and oral breathing while they were awake. Three-dimensional reconstructed stereolithography models and digital unstructured grid models were created and airflow simulations were performed using computational fluid dynamics software.

### Results

Airflow velocity was significantly higher during oral breathing than during nasal breathing with open or closed mouth. No significant difference in maximum velocity was noted between nasal breathing with closed and open mouth. However, airflow during nasal breathing with open mouth was slow but rapidly sped up at the lower level of the velopharynx, and then spread and became a disturbed, unsteady stream. In contrast, airflow during nasal breathing with closed mouth gradually sped up at the oropharyngeal level without spreading or disturbance. Negative static pressure during oral breathing was significantly decreased; however, there were no significant differences between nasal breathing with closed or open mouth.

### Conclusions

Computational fluid dynamics results during nasal and oral breathing revealed that oral breathing is the primary condition leading to pharyngeal airway collapse based on the

**Competing interests:** The authors have declared that no competing interests exist.

concept of the Starling Resistor model. Airflow throughout the entirety of the breathing route was smoother during nasal breathing with closed mouth than that with open mouth.

## Introduction

Mouth opening and oral breathing during sleep are thought to be associated with narrowing of the pharyngeal lumen and decreases in retroglossal diameter, which increase upper airway collapsibility and may lead to airway obstruction. It has been reported that upper airway collapsibility and resistance during sleep are significantly higher in people who breath through the mouth than in those who breath through the nose, which is different from what is seen in the conscious state. Meurice et al. demonstrated that mouth opening increased upper airway collapsibility during sleep [1]. Fitzpatrick et al. confirmed that during sleep, upper airway resistance during oral breathing was 2.5 times higher than that during nasal breathing [2]. Ayuse et al. examined upper airway critical pressure (Pcrit) in closed mouths, mouths opened moderately, and mouths opened maximally during sedation [3]. They reported that maximal mouth opening increased Pcrit to $-3.6 \pm 2.9$ cmH$_2$O, whereas Pcrit in moderate mouth opening was $-7.2 \pm 4.1$ cmH$_2$O and Pcrit in closed mouths was $-8.7 \pm 2.8$ cmH$_2$O, suggesting that maximal mouth opening increases upper airway collapsibility, which contributes to upper airway obstruction.

Although several physiological studies have been reported, the aerodynamics of nasal and oral breathing remain unclear. The purpose of this study was to investigate the effect of breathing route on the collapsibility of the pharyngeal airway, represented by airflow velocity and static pressure calculated using computational fluid dynamics (CFD) technology, in patients with obstructive sleep apnea (OSA).

## Methods

### Participants

Participants were 14 Japanese men with OSA and no nasal obstruction (age, $42.6 \pm 7.7$ years; body mass index, $28.4 \pm 5.5$ kg/m$^2$; apnea–hypopnea index, $43.7 \pm 21.6$/h; nasal resistance, $0.27 \pm 0.11$ Pa/cm$^3$/s). The following procedures were conducted for all participants: standard type 1 in-laboratory overnight polysomnography (PSG) (Alice 6, Philips Respironics, Pittsburgh, PA) in accordance with the American Academy of Sleep Medicine (AASM) scoring manual ver. 2.5, [4] and total inspiratory nasal resistance (NR) at $-100$ Pa with an anterior rhinomanometer (HI-801, Chest M.I., Inc., Tokyo, Japan) in the supine position. Those with OSA had AHI $\geq 15$/h, and those without nasal obstruction had total nasal resistance $\leq 0.50$ Pa/cm$^3$/s. We measured volumetric flow rates in a steady breathing state as a substitute marker for ventilatory drive. We used a Fleisch pneumotachometer (Laminar Flow Meter LFM-317; Metabo, Lausanne, Switzerland) along with a pressure sensor during nasal breathing with closed mouth, nasal breathing with open mouth, and oral breathing.

### Computational fluid dynamics analyses

Computed tomography (CT) scans of the nose, sinuses, and pharynx were taken at 0.5-mm intervals (Toscaner-32251μhd; Toshiba IT & Control Systems, Tokyo, Japan) during nasal breathing with closed mouth, nasal breathing with open mouth, and oral breathing while the participants were awake. We controlled each participant's breathing in a steady state with

volumetric flow rates. Individual three-dimensional (3D) reconstructed stereolithography (STL) models were created using image analysis software (Intage Volume Editor; Cybernet Systems, Ann Arbor, MI). These 3D reconstructed STL models included the nasal cavity, paranasal sinuses, pharynx, and soft tissue surrounding the airway (Fig 1). The digital unstructured grid models were meshed with 8 million hexahedral cells using the Intage Volume Editor and Hexpress/Hybrid (Numeca International, Brussels, Belgium). Geometrical modeling from medical image data and CFD analyses were conducted using a methodology described in our previous study [5, 6]. In brief, the surfaces were highly corrugated due to artifacts of digitization and were therefore smoothed to facilitate computational meshing. For inspiratory flow CFD analysis, the inlet boundary was set at a cross-section of the nostrils and an outlet boundary was set at a cross-section of the bottom of the hypopharynx. Inlet boundary conditions were set with atmospheric pressure conditions, and the inlet velocity distributions were approximated as flat, neglecting the boundary layer. The outlet boundary conditions were set with static pressures that corresponded to the volume flow conditions for the current cases. For expiratory flow analysis, these conditions were reversed.

Airflow simulations were conducted using Navier–Stokes equations in CFD software (Fine/Open, ver. 2.10.4; Numeca, Brussels, Belgium). Simulations were run over a 24-hour period on a 64-bit workstation with 24 GB of memory and 6 CPUs. Atmospheric pressure at 20˚C was applied to the inlet boundary (101.325 kPa = 1033.26 cmH$_2$O), with volumetric flow rates for inspiration and expiration of 320 mL/s in cases with nasal breathing, 45 mL/s in cases with oral breathing at the nostrils, and 560 mL/s in cases with oral breathing in front of the mouth. Air density was 1.204 kg/m$^3$. Air mass flow rate was $3.853 \times 10^{-4}$ kg/s in cases with nasal breathing and $7.285 \times 10^{-4}$ kg/s in cases with oral breathing. Nasal wall boundary conditions were heat-insulated walls with viscosity and turbulence taken into consideration. A no-slip boundary condition was applied on all nasal airway surfaces. Simulation models were confirmed to agree with measured airflow values.

All calculations were steady-state calculations using the maximum instantaneous flow rates measured during inspiration. The averaged Reynolds number in this study was around 3500. The Spalart-Allmaras one-equation turbulence model was used with the extended wall function for all cases in this study. The inlet turbulence boundary conditions were set with turbulence viscosity of 0.0001 m$^2$/s in our empirical models. The convergence of the CFD calculations was determined on the assumption that the average residual of CFD iterations should be less than $10^{-6}$ or the mass flow rate difference between inlet and outlet boundaries should be less than 0.5%.

Airflow velocity, wall shear stress, and static pressure in the nasal cavities and pharynx were analyzed in patients with OSA during nasal breathing with closed mouth, nasal breathing with open mouth, and oral breathing.

### Ethics and statistics

The Ethics Committee of Teikyo University approved this study (approval number 14–063) and written informed consent was obtained from all participants.

All descriptive statistics calculated for each variable are presented as the mean ± standard deviation. Variables were evaluated by one-way analysis of variance (ANOVA) among the three breathing conditions. A *p* value less than 0.01 was considered to indicate statistical significance. For multiple comparisons (*post hoc* test), variables were analyzed using the Bonferroni test. For comparisons between two conditions, variables were evaluated by Wilcoxon signed-rank test. Difference in airflow volume between inspiration and expiration was analyzed using

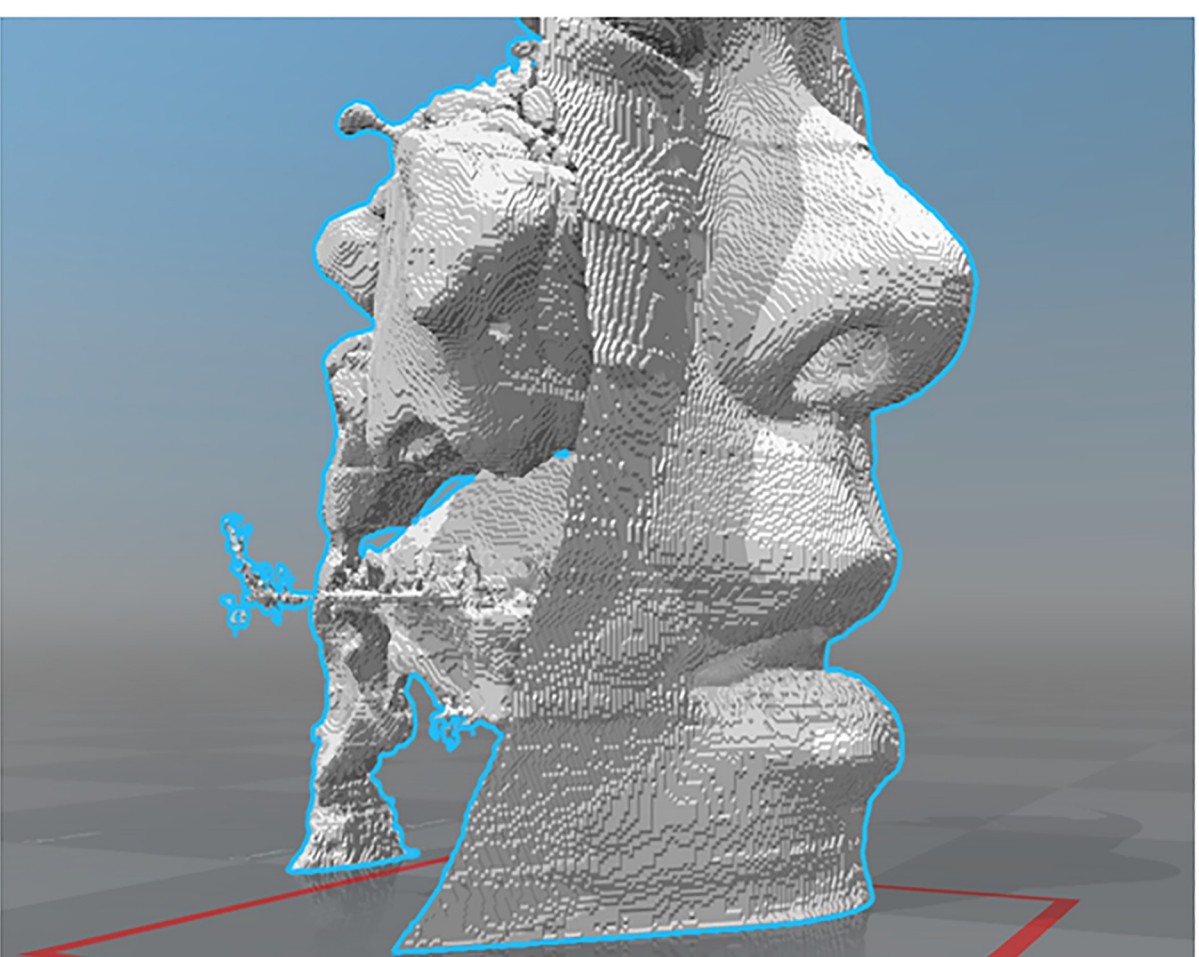

**Fig 1. 3D reconstructed STL model.**

a 2 × 2 Chi-square test. A $p$ value less than 0.05 was considered to indicate statistical significance.

## Results

The STL models revealed that the tongue touched the hard palate during nasal breathing with closed mouth, whereas a low tongue position that did not touch the hard palate was observed during nasal and oral breathing with open mouth.

The inspiratory airflow velocity was higher during oral breathing (as high as 9.37 ± 1.07 mL/s) than during nasal breathing with open or closed mouth ($p$ = 0.04) (Fig 2, Table 1). No significant difference was noted between nasal breathing with closed mouth (8.30 ± 1.07 mL/s) and nasal breathing with open mouth (7.93 ± 1.16 mL/s) (Figs 2 and 3, Table 1). During nasal breathing with open mouth, the inspiratory airflow in the nasal cavity and pharynx was relatively slow; it rapidly sped up at the lower level of the velopharynx, the junction of the nasal and oral breathing routes, then spread and became a disturbed, unsteady stream (Figs 2 and 3). A small amount of stream flowed into the mouth, and certain components of the inhaled air passed through the ostia into the maxillary sinuses before moving to the pharynx (Figs 2 and 3). In contrast, during nasal breathing with closed mouth, the inspiratory airflow was

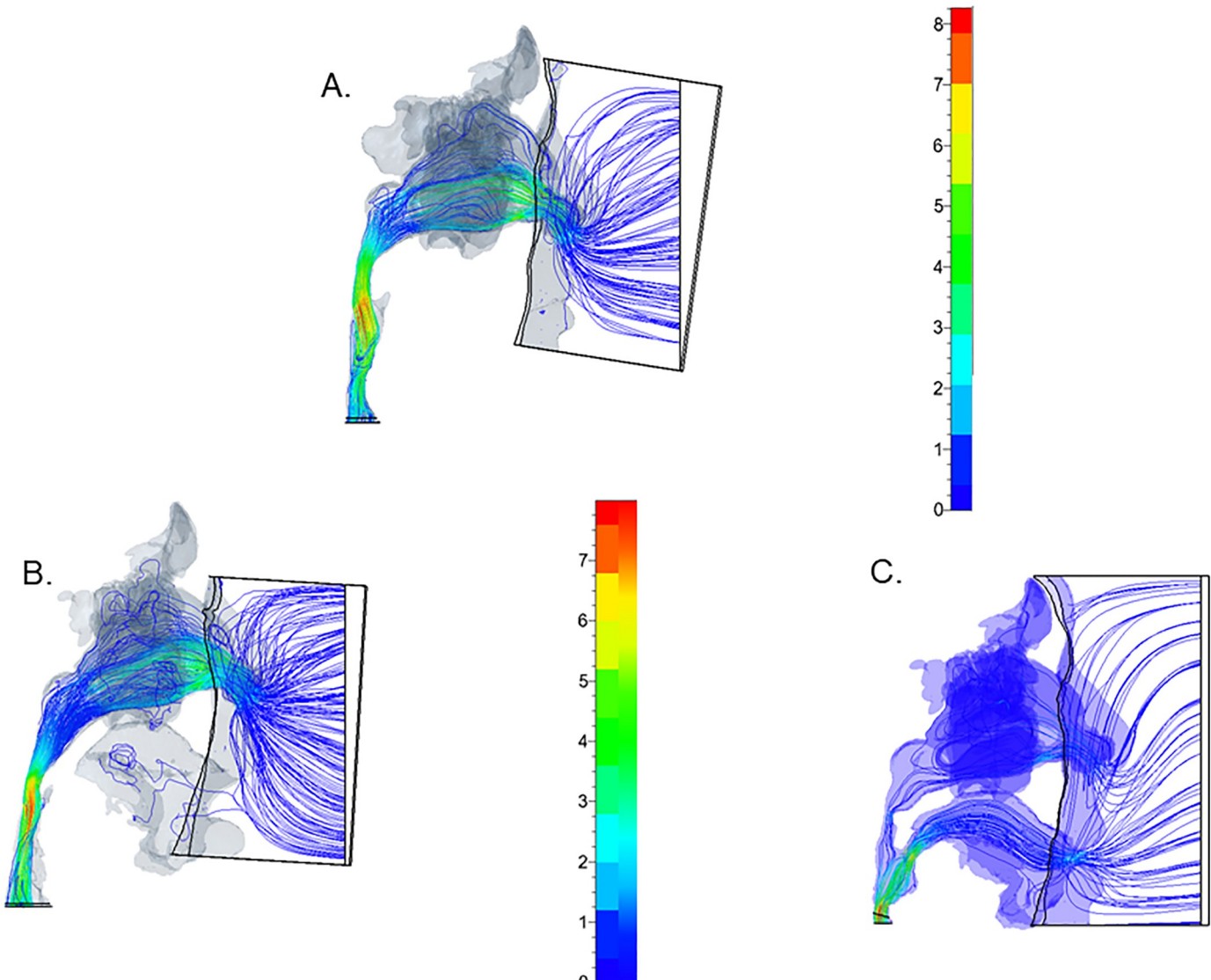

**Fig 2. Airflow imaging and velocity contours during inspiration, side view.** (A) Nasal breathing with closed mouth, (B) Nasal breathing with open mouth, (C) Oral breathing.

smooth throughout the breathing route, without spreading, disturbance, or instability; it gradually sped up to the maximum velocity at the oropharyngeal level without flowing into the oral cavity or the maxillary sinuses (Figs 2 and 3). CFD analyses showed that patients breathed 100% via the nose both in inspiration and expiration during nasal breathing, even with open mouth (Figs 2 and 4). In contrast, during oral breathing, patients breathed 28.7% ± 3.3% via the nose and 71.3 ± 5.1% via the mouth during inspiration, and breathed 20.4% ± 2.5% via the nose and 79.6% ± 4.4% via the mouth during expiration ($p = 0.17$) (Figs 2 and 4).

Next, the wall shear stress and static pressure distribution during inspiration were calculated. Wall shear stress during nasal breathing with closed mouth was 2.18 ± 0.37 Pa, and during nasal breathing with open mouth was 2.41 ± 0.28 Pa, showing a significant difference ($p = 0.024$) (Fig 5, Table 1). We were unable to perform CFD analyses of the wall shear stress

**Table 1. Summary of CFD results.**

| | A: nasal breathing with closed mouth | B: nasal breathing with open mouth | C: oral breathing | |
|---|---|---|---|---|
| Inspiratory airflow velocity (mL/s) (95% confidence interval) | 8.30 ± 1.07 (7.69 to 8.92) | 7.93 ± 1.16 (7.27 to 8.60) | 9.37 ± 1.07 (8.75 to 9.98) | * $p = 0.04$ † $p = 0.04$ ‡ $p = 0.04$ # N.S. |
| Negative static pressure (Pa) (95% confidence interval) | −66.2 ± 7.55 (−61.8 to −70.5) | −58.2 ± 7.97 (−53.6 to −62.8) | −121.8 ± 13.9 (−113.7 to −129.8) | * $p < 0.01$ † $p < 0.01$ ‡ $p < 0.01$ # N.S. |
| Wall shear stress (Pa) (95% confidence interval) | 2.18 ± 0.37 (1.97 to 2.38) | 2.41 ± 0.28 (2.26 to 2.57) | Unparsable | # $p = 0.024$ |

* among the three conditions (ANOVA test);

† between nasal breathing with closed mouth and oral breathing (Bonferroni test);

‡ between nasal breathing with open mouth and oral breathing (Bonferroni test);

# between nasal breathing with closed and open mouth (Bonferroni test among the three conditions, Wilcoxon signed-rank test between the two conditions)

during oral breathing, because we found it difficult to analyze wall shear stress during inspiration at the junction of nasal and oral flow.

Negative static pressure was also similar during nasal breathing with closed mouth (−66.2 ± 7.55 Pa) and nasal breathing with open mouth (−58.2 ± 7.97 Pa), showing no significant difference (Fig 6, Table 1). In contrast, negative static pressure during oral breathing decreased the most significantly, down to −121.8 ± 13.9 Pa ($p < 0.01$) (Fig 6). CFD results are summarized in Table 1.

## Discussion

Two novel findings were obtained in this CFD study. First, airflow velocity and static pressure were highest during oral breathing, suggesting that oral breathing is the primary condition leading to pharyngeal collapse in the three breathing conditions. Second, the airflow during nasal breathing with closed mouth was smooth throughout the whole breathing route—without spreading, disturbance, or instability—whereas that during nasal breathing with open mouth became a spreading and disturbed, unsteady stream.

Although several physiological studies have shown that mouth opening increases upper airway collapsibility, no reports to date have used CFD technology to investigate the effect of nasal and oral breathing route on upper airway collapsibility. Some studies, including our previous study, have evaluated the efficacy of OSA treatments by using CFD simulations with parameters such as velocity and static pressure; improvements in these CFD parameters after OSA treatment have been reported [6–8]. This is the first report to use CFD technology for investigating the effect of nasal and oral breathing route on upper airway collapsibility.

### Airflow velocity, static pressure, and collapsibility

Maintenance of upper airway patency is a primary physiologic condition during sleep; failure leads to collapse of the upper airway. Dynamic alterations in patency in patients with OSA are modeled as a function of transmural pressure across collapsible segments. Collapsibility of the upper airway is based on the Starling resistor model, a theoretical model related to Bernoulli's theory, whereby maximal airflow through the collapsible segment is dependent on the resistance of the upstream and downstream rigid segment and the pressure surrounding the collapsible segment [9–10]. In this model, the upper airway is considered to contain a

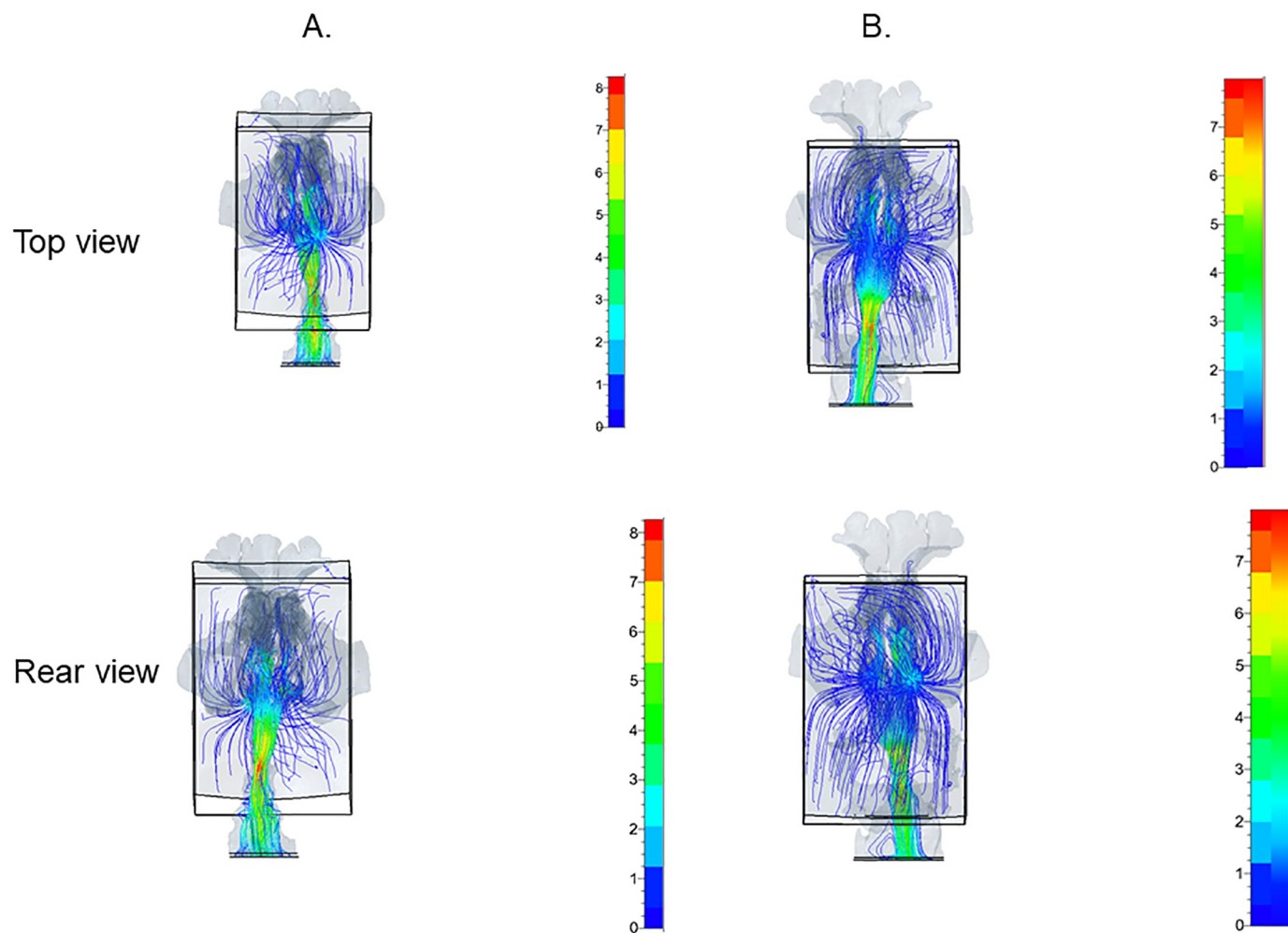

**Fig 3. Airflow imaging and velocity contours during inspiration, top and rear view.** (A) Nasal breathing with closed mouth, (B) Nasal breathing with open mouth.

compressible segment with a smaller cross-sectional area than the two rigid segments of the upper airway that it connects, so that the airflow velocity is greater through it than through the rigid segments. When the upstream and downstream pressures are lower than the critical pressure surrounding the collapsible segment, the negative intraluminal pressure (negative static pressure) decreases and the velocity of inspiratory airflow increases. Thus, obstruction occurs, the airway closes, and airflow ceases. This model postulates the oropharynx as a collapsible segment. Static pressure changes are amplified dynamically in this segment via the Bernoulli effect, and the airflow velocity through the upper airway is proportional to the static pressure gradient across the entire airway.

Based on the concept of the Starling resistor model, collapsibility is dependent on the airflow velocity and the static pressure through the oropharynx. Detailed values for airflow velocity and static pressure through the pharyngeal airway can be calculated using CFD. Our results showed that airflow velocity and static pressure were significantly increased during oral breathing, indicating that airflows with different velocities merged to generate friction and swirl, which led to loss of pressure and an increase in entropy that facilitated collapse.

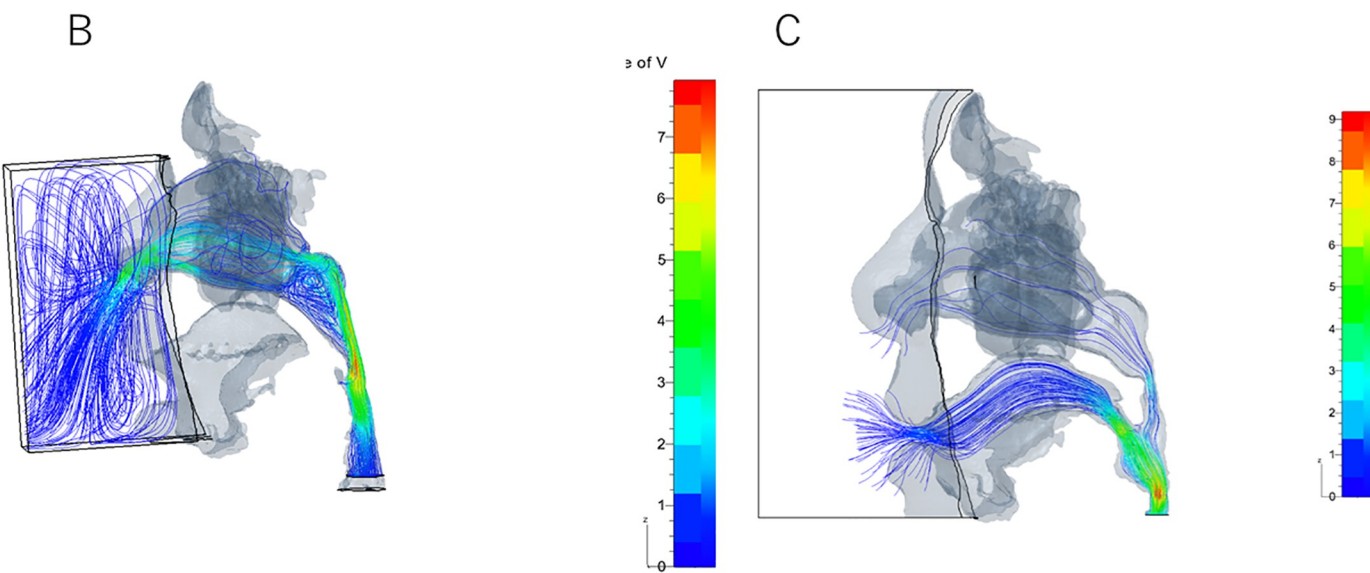

**Fig 4. Airflow imaging and velocity contours during expiration, side view.** (B) Nasal breathing with open mouth, (C) Oral breathing.

## Oral breathing, nasal obstruction, and pharyngeal collapse

During sleep, the physiology of the upper and lower airways and respiratory control encourage nasal breathing rather than oral breathing. However, in nasal diseases such as nasal septum deviation or inferior turbinate hypertrophy, nasal obstruction can be bypassed by opening the mouth and allowing a greater volume of air to be inspired and expired. McLean et al. showed that oral breathing during sleep is induced by increased nasal resistance [11]. Our previous study showed that oral flow can be divided into three main patterns [12]. In these three patterns, spontaneous arousal-related oral flow was associated with nasal obstruction, typically seen in patients with mild to moderate sleep-disordered breathing. Increased nasal resistance leads to mouth opening and oral breathing. If nasal airway obstruction is severe, with high

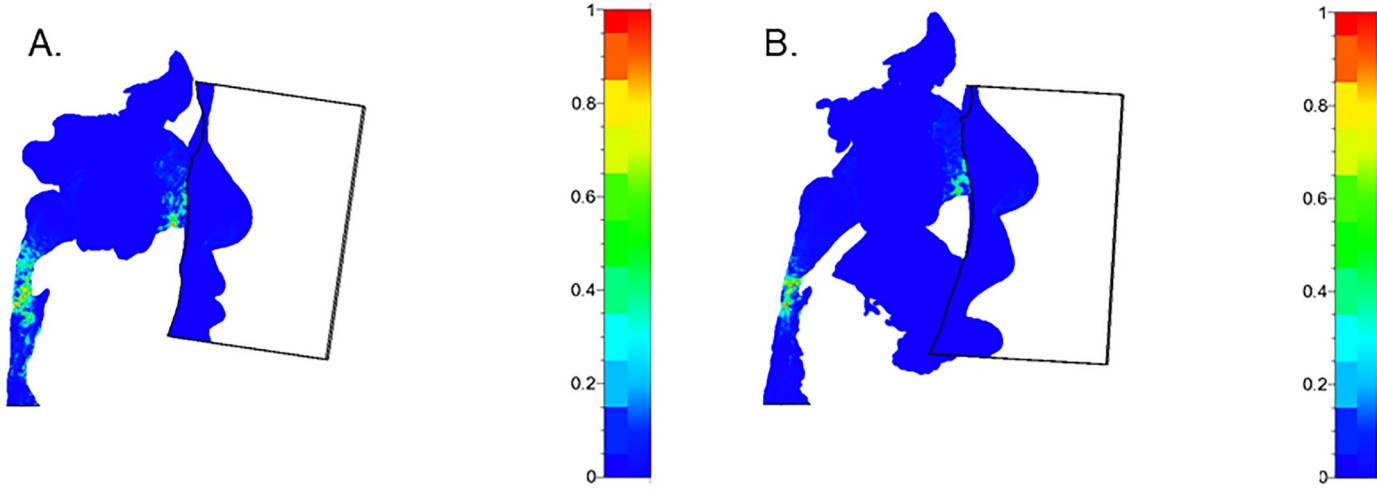

**Fig 5. Wall shear stress distribution during inspiration.** (A) Nasal breathing with closed mouth, (B) Nasal breathing with open mouth.

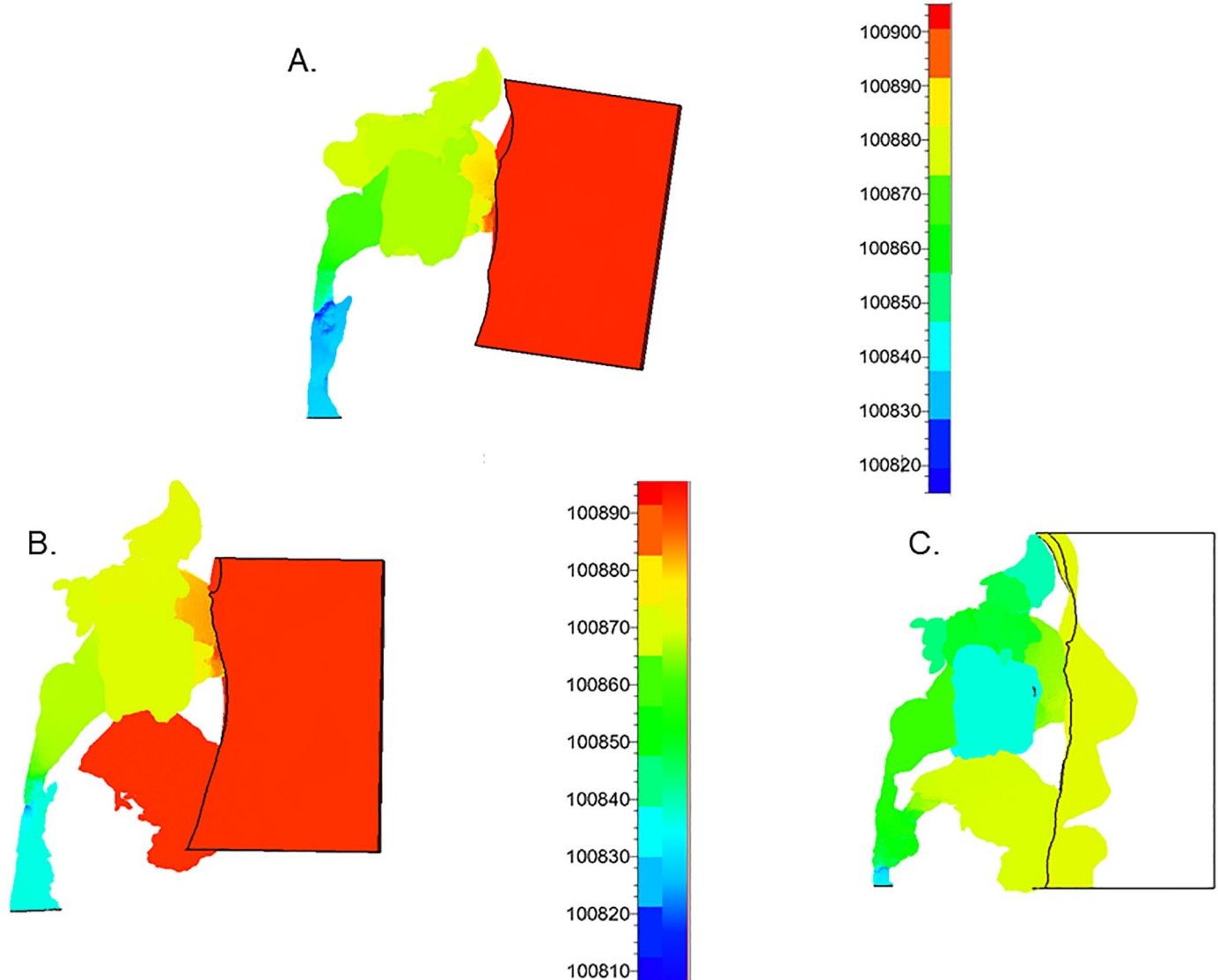

**Fig 6. Static pressure distribution during inspiration.** (A) Nasal breathing with closed mouth, (B) Nasal breathing with open mouth, (C) Oral breathing.

inspiratory resistive loads, nasal resistance exceeds a certain threshold and nasal breathing switches to oral breathing to bypass nasal airway obstruction. The results of this study are in agreement with those of the above-mentioned physiological studies. In patients susceptible to airway collapse or habitual oral breathing, oral breathing leads to mouth opening and sustained oral breathing; consequently, post-event or during-event oral flows would occur and induce respiratory events such as apnea or hypopnea.

## Study limitations

A limitation of this study is that, first, we were unable to perform CFD analyses of the wall shear stress during oral breathing with open mouth, because of computational difficulties in analyzing wall shear stress during inspiration at the junction of nasal and oral flow using the CFD software. Further research to confirm the accuracy of analyses of wall shear stress at the

junction are needed. Second, CT scans were performed while patients were conscious. It was difficult to distinguish nasal breathing with open mouth from nasal breathing with closed mouth and oral breathing during sleep under CT scanning conditions; thus, patients could have breathed with different amounts of effort, resulting in a large bias.

## Opening the mouth and opening the ostium

Simulation models of the paranasal sinuses have been reported, showing increases of the airflow into the maxillary sinuses after nasal surgeries [13–14]. We showed that increased airflow streamlines passed into the maxillary sinuses during nasal breathing with open mouth compared to nasal breathing with closed mouth. When we open the mouth, the palatal tensor muscle makes the eustachian tube open, and airflow goes into the eustachian tube. However, there is no muscle opening the ostium of the maxillary sinus when the mouth opens. The relationship between opening the mouth and opening the ostium of the maxillary sinus is unknown. This could provide rhinologists with an interesting perspective; further investigations are required.

## CFD in the upper airway

The aerodynamics of the nose and airway are complex due to their geometry and wall conditions. Static pressure is considered the key to elucidating pharyngeal collapsibility in patients with OSA. However, our results showed that airflow imaging and velocity contours provided detailed aerodynamics of nasal and oral breathing, demonstrating that airflow imaging is also an essential part of CFD analyses in the nose and airway in patients with OSA.

CFD studies to date have improved our understanding of pathogenesis on airflow and implications on nose and airway physiology. We could understand the pathogenesis of the nose and airway in greater detail by using these CFD assessments.

## Author Contributions

**Conceptualization:** Masaaki Suzuki.

**Formal analysis:** Masaaki Suzuki, Tadashi Tanuma.

**Funding acquisition:** Masaaki Suzuki.

**Investigation:** Masaaki Suzuki.

**Methodology:** Masaaki Suzuki, Tadashi Tanuma.

**Project administration:** Masaaki Suzuki.

**Writing – original draft:** Masaaki Suzuki.

**Writing – review & editing:** Masaaki Suzuki.

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
