## [Decision Letter · Decision Letter 0]

31 Dec 2019

PONE-D-19-30629

Comparison between nasal and oral breathing in patients with obstructive sleep apnea: computational fluid dynamics analyses

PLOS ONE

Dear Dr. Masaaki Suzuki,

Thank you for submitting your manuscript to PLOS ONE. After careful consideration, we feel that it has merit but does not fully meet PLOS ONE’s publication criteria as it currently stands. Therefore, we invite you to submit a revised version of the manuscript that addresses the points raised during the review process.

ACADEMIC EDITOR: 

As the reviewers' concern, the sample size is too small to make solid conclusion and  conditions of experiment should be clarified. Moreover, the illustration of procedure and data should be organized to help readers to understand better. I strongly recommend that  more subjects need to be recruited and the paper needs extensive revision to meet the standard of publication.

We would appreciate receiving your revised manuscript by Feb 14 2020 11:59PM. To enhance the reproducibility of your results, we recommend that if applicable you deposit your laboratory protocols in protocols.io, where a protocol can be assigned its own identifier (DOI) such that it can be cited independently in the future. For instructions see: http://journals.plos.org/plosone/s/submission-guidelines#loc-laboratory-protocols

We look forward to receiving your revised manuscript.

Kind regards,

Pei-Lin Lee, M.D., PhD

Academic Editor

PLOS ONE

2. We noticed you have some minor occurrence(s) of overlapping text with the following previous publication(s), which needs to be addressed:

https://doi.org/10.1371/journal.pone.0150951

In your revision ensure you cite all your sources (including your own works), and quote or rephrase any duplicated text outside the Methods section. Further consideration is dependent on these concerns being addressed.

4. Thank you for stating the following financial disclosure: "NO"

a)    Please provide an amended Funding Statement that declares *all* the funding or sources of support received during this specific study (whether external or internal to your organization) as detailed online in our guide for authors at http://journals.plos.org/plosone/s/submit-now.  

b)    Please state what role the funders took in the study.  If any authors received a salary from any of your funders, please state which authors and which funder. If the funders had no role, please state: "The funders had no role in study design, data collection and analysis, decision to publish, or preparation of the manuscript."

Reviewers' comments:

Reviewer's Responses to Questions

**Comments to the Author**

1. Is the manuscript technically sound, and do the data support the conclusions?

Reviewer #1: Yes

Reviewer #2: Partly

2. Has the statistical analysis been performed appropriately and rigorously? 

Reviewer #1: Yes

Reviewer #2: No

3. Have the authors made all data underlying the findings in their manuscript fully available?

Reviewer #1: Yes

Reviewer #2: No

4. Is the manuscript presented in an intelligible fashion and written in standard English?

Reviewer #1: Yes

Reviewer #2: No

5. Review Comments to the Author

Reviewer #1: In the current study, the authors investigated differences of airflow dynamics between three types of breathing, that is, nasal breathing with mouth closed, nasal breathing with mouth opened, and oral breathing, in OSA patients (n=6).

Their major finding includes nasal breathing with mouth closed showed the advantage in terms of upper airway collapsibility.

To this reviewer, the current study looks very interesting, and manuscript is well written and well discussed. However, some issues should be addressed for the acceptable form of publication.

Major comments:

Is there any way to identify or assess the anatomical features such as a position of tongue during three types of breathing? It is just curiosity of this reviewer.

The sample size is just 6 OSA patients, so suggestion would be that assessment of SDB characteristics and computational fluid dynamics in each patient should be performed if possible. Might be interesting.

Minor comment:

The authors looked at only inspiration phase. Did the authors instruct the patients how to exhale, i.e. breath out via mouth or nose, and so forth?

Reviewer #2: This study investigated the differences in parameters of computational fluid dynamics (CFD) between the nasal and oral breathing in patients with obstructive sleep apnea (OSA). According to preliminary data of six adulthood patients with OSA, the authors concluded that oral breathing is the primary condition leading to pharyngeal collapse and the airflow during nasal breathing with the closed mouth was smoother than that with open mouth. Although this study is interested, there are many issues need to be addressed. Please consider revising to improve the quality of this manuscript.

1. The study topic is vague. What did you want to compare? The topic should include the examination condition (conscious status or sleep status?)

2. The conclusions of the abstract are not supported by the results (There was no result showing oral breathing increased pharyngeal collapse.) or simply duplication (The airflow during nasal breathing with the closed mouth was smoother than that with open mouth.).

3. In the method section, please illustrate the procedures to control the respiratory force during nasal breathing without the mouth opening, nasal breathing with the mouth opening, and mouth breathing? If the respiratory forces were significantly different, measurements of CFD parameters (such as velocity) might be not objective.

4. Please provide several tables to summarize your results because the readers cannot track your data according to the figures.

5. How did you decide your sample size? Did you evaluate the data distribution (normal or not normal?) and the statistical power? Because the sample size of this study was relatively small, this study was very preliminary and hard to make a conclusion.

6. The discussion section was not well-written and effective. The authors should compare their findings with other studies to explain specific research outcomes. The authors may explain the reasons for these findings are specific to OSA. Please consider writing a more effective discussion.

7. During awake, conscious states, patients could breathe with different efforts resulting in a larger bias and did not mimic to sleep-disturbed breathing events. Please consider to add it to your limitation.

8. Please request for English editing of this manuscript. There were many grammar errors and confusing sentences in the manuscript.

6. PLOS authors have the option to publish the peer review history of their article (what does this mean?). If published, this will include your full peer review and any attached files.

Reviewer #1: No

Reviewer #2: Yes: Li-Ang Lee

---

## [Author Response · Author response to Decision Letter 0]

19 Feb 2020

Response to Reviewers

We would like to thank the reviewers for the latest review and apologize for delaying the resubmission of this revised manuscript. We have answered each of the points below. We increased the sample size and reorganized the manuscript to try to improve the logical flow of our argument. We hope that this more substantial reworking of the manuscript is satisfactory.

Reviewer #1: 

Major comments:

Is there any way to identify or assess the anatomical features such as a position of tongue during three types of breathing? It is just curiosity of this reviewer.

Thank you for pointing this out. The STL models revealed that the tongue touched the hard palate during nasal breathing with closed mouth, whereas a low tongue position that did not touch the hard palate was observed during nasal and oral breathing with open mouth.

We have added these findings to the Results section.

The sample size is just 6 OSA patients, so suggestion would be that assessment of SDB characteristics and computational fluid dynamics in each patient should be performed if possible. Might be interesting.

Thank you for pointing this out. Statistical power test suggested that sample needs more than 12.14 under the conditions of delta; 0.3, sd; 0.2, sig.level; 0.01, and power; 0.8. Therefore, we added eight patients to make sample size 14. Significant difference was observed between the nasal breathing with closed and open mouth in the wall shear stress, however, other statistical comparisons showed almost the same results as before.

We changed the Result section.

 Minor comment:

The authors looked at only inspiration phase. Did the authors instruct the patients how to exhale, i.e. breath out via mouth or nose, and so forth?

Thank you for pointing this out. During oral breathing, patients breathed 28.7% ± 3.3% via the nose and 71.3 ± 5.1% via the mouth during inspiration, and breathed 20.4% ± 2.5% via the nose and 79.6% ± 4.4% via the mouth during expiration (p= 0.17, 2x2 Chi square test). In contrast, patients breathed 100% via the nose both in inspiration and expiration during nasal breathing, even with open mouth. 

We have added these findings and CFD figures during expiration (Fig. 4) to the Results section.

Reviewer #2: 

1. The study topic is vague. What did you want to compare? The topic should include the examination condition (conscious status or sleep status?)

Thank you for pointing this out. The purpose of this study was to investigate the effect of breathing route on the collapsibility of the pharyngeal airway, represented by airflow velocity and static pressure calculated using CFD technology, in patients with OSA.

We changed the title, Introduction, and objective of the abstract to make this clear.

2. The conclusions of the abstract are not supported by the results (There was no result showing oral breathing increased pharyngeal collapse.) 

Thank you for pointing this out. We elucidated the pharyngeal airway collapsibility based on the concept of the Starling Resistor model. Collapsibility is dependent on the airflow velocity and the static pressure through the oropharynx. Detailed values for airﬂow velocity and static pressure through the pharyngeal airway can be calculated using CFD. Our results showed that airflow velocity and static pressure were significantly increased during oral breathing, indicating that airflows with different velocities merged to generate friction and swirl, which led to loss of pressure and an increase in entropy that facilitated collapse. 

We changed the Abstract and have added more detailed explanation of collapsibility to the Discussion section.

simply duplication (The airflow during nasal breathing with the closed mouth was smoother than that with open mouth.).

We have avoided the duplication in the Abstract.

3. In the method section, please illustrate the procedures to control the respiratory force during nasal breathing without the mouth opening, nasal breathing with the mouth opening, and mouth breathing? If the respiratory forces were significantly different, measurements of CFD parameters (such as velocity) might be not objective.

Thank you for pointing this out. We measured volumetric flow rates in a steady breathing state as a substitute marker for ventilatory drive using a pneumotachometer along with a pressure sensor rather than performing diaphragm electromyography during nasal breathing with closed mouth, nasal breathing with open mouth, and oral breathing. Simulation models were confirmed to agree with these measured values.

We have added this explanation to the Methods.

4. Please provide several tables to summarize your results because the readers cannot track your data according to the figures.

Thank you for pointing this out. We have added a summarized table of CFD results (Table 1).

5. How did you decide your sample size? Did you evaluate the data distribution (normal or not normal?) and the statistical power? Because the sample size of this study was relatively small, this study was very preliminary and hard to make a conclusion.

Thank you for pointing this out. Statistical power test suggested that sample needs more than 12.14 under the conditions of delta; 0.3, sd; 0.2, sig.level; 0.01, and power; 0.8. Therefore, we added eight patients to make sample size 14. Significant difference was observed between the nasal breathing with closed and open mouth in the wall shear stress, however, other statistical comparisons showed almost the same results as before.

We changed the Result section.

6. The discussion section was not well-written and effective. The authors should compare their findings with other studies to explain specific research outcomes. The authors may explain the reasons for these findings are specific to OSA. Please consider writing a more effective discussion.

Thank you for pointing this out. We have reorganized the Discussion to try to improve the logical flow of our argument. We have added essential discussion concerning velocity, pressure and collapsibility to the Discussion section, and deleted non-effective sections.

7. During awake, conscious states, patients could breathe with different efforts resulting in a larger bias and did not mimic to sleep-disturbed breathing events. Please consider to add it to your limitation.

Thank you for pointing this out. CT scans were performed while patients were conscious; thus, patients could have breathed with different amounts of effort, resulting in a large bias.

We have added this limitation section in the Discussion.

8. Please request for English editing of this manuscript. There were many grammar errors and confusing sentences in the manuscript.

The paper has been checked by a native English-speaking medical editor.

---

## [Decision Letter · Decision Letter 1]

20 Mar 2020

The effect of nasal and oral breathing on airway collapsibility in patients with obstructive sleep apnea: computational fluid dynamics analyses

PONE-D-19-30629R1

Dear Dr. Masaaki Suzuki

We are pleased to inform you that your manuscript has been judged scientifically suitable for publication and will be formally accepted for publication once it complies with all outstanding technical requirements.

With kind regards,

Pei-Lin Lee, M.D., PhD

Academic Editor

PLOS ONE

Additional Editor Comments (optional):

Both reviewers comment that all comments have been addressed.

Reviewers' comments:

Reviewer's Responses to Questions

**Comments to the Author**

1. If the authors have adequately addressed your comments raised in a previous round of review and you feel that this manuscript is now acceptable for publication, you may indicate that here to bypass the “Comments to the Author” section, enter your conflict of interest statement in the “Confidential to Editor” section, and submit your "Accept" recommendation.

Reviewer #1: All comments have been addressed

Reviewer #2: All comments have been addressed

2. Is the manuscript technically sound, and do the data support the conclusions?

Reviewer #1: Yes

Reviewer #2: Yes

3. Has the statistical analysis been performed appropriately and rigorously? 

Reviewer #1: Yes

Reviewer #2: Yes

4. Have the authors made all data underlying the findings in their manuscript fully available?

Reviewer #1: Yes

Reviewer #2: Yes

5. Is the manuscript presented in an intelligible fashion and written in standard English?

Reviewer #1: Yes

Reviewer #2: Yes

6. Review Comments to the Author

Reviewer #1: This review paper is a re-submission, and addresses the differences in pharyngeal airway collapsibility between nasal and oral breathing routs.

Overall the paper is improved over the initial submission, and my concerns have been adequately addressed.

Reviewer #2: Dear Dr. Suzuki,

Thanks for your revised manuscript. I find that my previous issues have been significantly addressed. This manuscript is more friendly to the audience and provides a reliable methodology to the peers. Your findings are interesting and provide a deep insight into nasal/oral breathing in obstructive sleep apnea. I have no further comments. Congratulate!

7. PLOS authors have the option to publish the peer review history of their article (what does this mean?). If published, this will include your full peer review and any attached files.

Reviewer #1: No

Reviewer #2: Yes: Li-Ang Lee

---

## [Editor Report · Acceptance letter]

30 Mar 2020

PONE-D-19-30629R1 

The effect of nasal and oral breathing on airway collapsibility in patients with obstructive sleep apnea: computational fluid dynamics analyses 

Dear Dr. Suzuki:

I am pleased to inform you that your manuscript has been deemed suitable for publication in PLOS ONE. Congratulations! Your manuscript is now with our production department. 

With kind regards,

on behalf of

Dr. Pei-Lin Lee 

Academic Editor

PLOS ONE